# Synthesis and Evaluation of the Antitumor Activity of Novel 1-(4-Substituted phenyl)-2-ethyl Imidazole Apoptosis Inducers In Vitro

**DOI:** 10.3390/molecules25184293

**Published:** 2020-09-18

**Authors:** Zhen-Wang Li, Chun-Yan Zhong, Xiao-Ran Wang, Shi-Nian Li, Chun-Yuan Pan, Xin Wang, Xian-Yu Sun

**Affiliations:** 1College of Animal Science and Technique, Heilongjiang Bayi Agriculture University, Daqing 163319, Heilongjiang, China; lzwlizhenwang@163.com (Z.-W.L.); 18345976963@163.com (C.-Y.Z.); wxr946909131@163.com (X.-R.W.); 13936989232@163.com (S.-N.L.); pcy0459@163.com (C.-Y.P.); wangxin7777@vip.163.com (X.W.); 2School of Biology and Food Engineering, Changshu Institute of Technology, Changshu 215500, Jiangsu, China

**Keywords:** imidazole, antitumor, apoptosis, Bcl-2, Bax

## Abstract

Novel imidazole derivatives were designed, prepared, and evaluated in vitro for antitumor activity. The majority of the tested derivatives showed improved antiproliferative activity compared to the positive control drugs 5-FU and MTX. Among them, compound **4f** exhibited outstanding antiproliferative activity against three cancer cell lines and was considerably more potent than both 5-FU and MTX. In particular, the selectivity index indicated that the tolerance of normal L-02 cells to **4f** was 23–46-fold higher than that of tumor cells. This selectivity was significantly higher than that exhibited by the positive control drugs. Furthermore, compound **4f** induced cell apoptosis by increasing the protein expression levels of Bax and decreasing those of Bcl-2 in a time-dependent manner. Therefore, **4f** could be a potential candidate for the development of a novel antitumor agent.

## 1. Introduction

The essential hallmarks of cancer include an altered cancer–cell intrinsic metabolism, i.e., tumor cell metabolism, which was seen as cancer’s Achilles heel [1]. Targeting apoptosis, a mechanism of programmed cell death, has been one of the most successful therapeutic approaches in cancer therapy. Under certain physiological or pathological conditions, cells can protect themselves from undergoing division and death. Apoptosis is a spontaneous and active process of cell death that commonly exists in multicellular organisms as an inherent mechanism for maintaining a relatively stable number of cells [2]. It is generally accepted that tumors are caused by a low rate of apoptosis and excessive proliferation. Any abnormalities in the rate of apoptosis can cause tumors and other diseases. If the proliferation of tumor cells can be stopped or if apoptosis can be induced, tumors may stop growing [3]. This application of apoptosis offers a broad prospect for the integration of cell biology, molecular biology, and many other disciplines related to the subject of the “life system”. Since Hickman proposed that inducing the apoptosis of tumor cells could be paramount for tumor treatment, treating cancer via therapeutic induction of apoptosis has gradually become a popular area of research [4].

To identify new apoptosis-inducing candidates, we attempted to modify the structure of imidazole and its analogs. Imidazole and the related azole moiety are significant pharmacophores in various clinical drugs including antitumor agents [5,6,7,8,9,10,11,12,13,14]. Nutlins-1, which has imidazole as a pharmacophore, displaces recombinant p53 protein from its complex with MDM2 and selectively inhibits MDM2-p53 interaction both in vitro and in vivo [15]. The antitumor potency of Tanshinone-IIA has been enhanced by the introduction of an imidazole moiety (Figure 1, TA12) in order to block cancer cell invasion and metastasis in vivo [16]. *N*-phenylimidazole (NMIB) possesses outstanding cytotoxic activity [17]. Previously, we reported a series of imidazole-fused quinoline compound **I** and its open-chain analogue compound **II**, 2-ethyl-1-phenyl-1*H*-imidazole (Figure 2) [18,19]. In the preliminary work of this paper, we found that compound **I** possesses weak antitumor activity in vitro, and compound **II** showed a little stronger activity against lung and liver cancer cells. We speculated that its flexible structure caused this difference in antitumor activity. However, both their antitumor activities were not strong enough as a potent antitumor agent. So, using compound **II** as a lead, we designed and synthesized several structurally modified derivatives. The effects of Log*P* and hydrogen bonds on bioavailability were considered, and several aliphatic and aromatic amines were selected and linked to the 1-phenyl imidazole scaffold using a succinyl group as a cross-linker. This approach yielded several novel imidazole derivatives with cLog*P* values ranging from 1.99 to 4.34. Two peptide bonds present in these structures may be good hydrogen bond receptors, and we expected that these compounds would exhibit interesting pharmacological properties.

Here, we report the synthesis and proapoptotic activity of novel 1-(4-substituted phenyl)-2-ethyl imidazole derivatives that exhibited excellent antitumor activity. We explored the structure–anticancer activity relationships and antitumor mechanisms of these compounds from the perspective of apoptosis.

## 2. Results and Discussion

### 2.1. Chemistry

The synthesis of the designed imidazole derivatives **4a**–**4s** is outlined in Scheme 1. Compound **1** was prepared by 2-ethylimidazole with 4-fluoronitrobenzene chloride in a NaOH solution. Compound **2** was obtained by reducing compound **1** with hydrogen catalyzed by Pd/C. Treatment of compound **2** with succinic anhydride resulted in compound **3**. Derivatives **4a**–**4s** were prepared by reacting compound **3** and various organic amines (containing *n*-alkane amine, substituted aniline, substituted benzylamine, 2-aminothiazole, and 2-picolylamine) with EDCI and DMAP in DMF. The yields of compounds **4a**–**4s** were moderate ranging from 27 to 62%, which is a little bit lower than would be expected. Actually, there were some impurities in the reaction mixture. Additionally, also recrystallization caused some target compounds loss. The prepared compounds were identified by ^1^H-NMR, ^13^C-NMR, and HRMS (Appendix A).

### 2.2. In Vitro Antiproliferative Activity

The synthesized compounds were subjected to a cytotoxicity assay using the MTT method, as described previously. IC_50_ is defined as the drug concentration causing a 50% reduction in cell numbers compared to the untreated control. For screening, we evaluated the effects of the compounds on the growth of three cancer cell lines (A549, SGC-7901, and HeLa). In addition, their toxicities were tested using one normal cell line (L-02). Their anticancer activities were measured and are presented in Table 1. The results are the averages of four separate measurements. MTX and 5-FU served as positive controls.

As shown in Table 1, 19 derivatives containing alkyl, benzyl, and phenyl groups and exhibiting different space effects and lipid–water partition coefficients were designed. Most of the tested compounds exhibited appreciable anticancer activity; the inhibitory activity of several compounds was better than that of 5-fluorouracil (5-FU) and methotrexate (MTX), which were used as positive controls. 5-FU and MTX target thymidylate synthase and dihydrofolate reductase (DHFR), respectively, and are used extensively to treat cancer.

We calculated the LogP for each target compound; a drug needs to have a suitable cLog*P* value as well as good activity for its utility. As shown in Table 1, compound **4f** appeared to be the most promising structure, with a cLog*P* value of 4.34.

By testing the inhibitory activity of 19 compounds using three cancer cell lines, we found that the aliphatic-substituted compounds were less active than the aromatic-substituted compounds. Among the fat-substituted compounds (**4a**–**4c**), **4c** was the most active in this series, with an IC_50_ value of 18.53 µM, which is comparable to that of 5-FU. Analysis of the inhibitory activity of compounds **4a**–**4c** against three cancer cell lines showed that the length of the alkyl chain had a direct impact on the antitumor activity of the imidazole derivative. We observed that the IC_50_ gradually increased with the increasing alkyl chain length, thus culminating with compound **4c** having an amyl substituent.

All of the phenyl-substituted compounds, **4d**–**4j**, exhibited comparable or higher activities than the positive controls, except compounds **4d** and **4j**. The pharmacological analyses showed that the Br substituent contributed more to antitumor activity than Cl or F. *p*-Br compound **4f**, with IC_50_ of 6.60, 3.24, and 5.37 µM against A549, HeLa, and SGC-7901, was more active than the other halogen-containing compounds. This compound was approximately five times more potent than MTX in the in vitro evaluation of cancer cell lines. Interestingly, *p*-NH_2_ compound **4j** was approximately twice as potent as MTX against HeLa cells, with an IC_50_ value of 7.42 µM; however, weak inhibitory activity was observed with the A549 and SGC-7901 cell lines.

The analysis of seven benzyl-containing compounds **4k**–**4q** showed that most compounds containing the substituted benzyl rings possessed stronger antitumor activity (10.96–75.87 µM) compared to that of compound **4k**, which had a non-substituted benzyl ring. *p*-Cl derivative **4l** exhibited much stronger antitumor activity in the A549 cell line than the positive-control drugs. Compound **4m** showed the strongest inhibition potency in this series with an IC_50_ value of 4.07 µM against HeLa cells. Compound **4q** showed the strongest inhibition potency in this series with an IC_50_ value of 2.96 µM against SGC-7901 cells, which was approximately five-fold stronger than the MTX control. Compound **4n** showed the same activity trend as compound **4j**, and both displayed stronger inhibitory activity in HeLa than in the A549 and SGC-7901 cells. This indicates that HeLa cells are more sensitive to the test compounds than the other two cell lines.

The pharmacological results of compound **4r** indicate that compared to the activity of **4k**, the contribution of pyridine to the activity is not as effective as that of the benzene ring. Compound **4s**, a thiazole-containing compound, showed similar activity to the control drugs in the A549 and HeLa cells.

Overall, most of the tested compounds exhibited significantly stronger antitumor activities than compound **I** and leading compound **II**, the presence of phenyl or benzyl groups linked by the succinyl group greatly enhanced the antitumor activity of imidazole derivatives, indicating that our original design is both feasible and effective. The results illustrate that there are no significant differences between the effects of electron-donating and electron-withdrawing substituents in this series. We identified compound **4f** as a potent agent against cancer cells that was several times more effective than the clinical drugs 5-FU and MTX.

### 2.3. Selective Inhibition of Cancer Cell Growth by Compounds ***4c***, ***4e***–***4i***, ***4l***, ***4m***, and ***4q*** In Vitro

The lack of selectivity between normal and cancer cells is one of the primary limitations of antitumor drugs [20]. We evaluated the cytotoxicity of compounds **4c**, **4e**–**4i**, **4l**, **4m**, and **4q** in the normal cell line L-02 to determine the selectivity index (defined as the ratio of cytotoxicity in L-02 cells compared to that in cancer cells). As shown in Table 2, the selectivity index indicated that the tolerance of normal L-02 cells to **4f** was better than that of tumor cells with a range of 23–46-fold, and this selectivity was significantly higher than that of 5-FU and MTX. Compound **4q** yielded satisfactory SI values in HeLa and SGC-7901 cells but an unsatisfactory SI value in A549 cells. Compound **4f** not only exhibited outstanding antiproliferative activity in the three cancer cell lines but also selectively inhibited tumor cells. Compound **4f** was selected for the subsequent biological studies.

### 2.4. Compound ***4f*** Induced Apoptosis in HeLa Cells

To determine whether imidazole derivatives induce cancer cell apoptosis, the most active compound **4f** (3.24 µM) was selected to performed Hoechst/PI double staining testing with HeLa cells to elucidate the underlying mechanism. As depicted in Figure 3A, the experimental group was significantly better than the control treatment with 5-FU. In particular, at 24 h, **4f** treatment resulted in a 68.2% apoptosis rate compared to the 39.6% rate induced by 5-FU (Figure 3B).

### 2.5. Western Blot of Compound ***4f*** in HeLa Cells

The occurrence of cancer is related to various factors and the mechanism underlying its induction is complex and diverse. The inhibition of oncogenesis and metastasis is primarily regulated by apoptosis induction. The expression of some apoptotic proteins is altered during apoptosis [21,22]. We examined whether the anticancer mechanism of **4f** is related to apoptosis and monitored the expression of four peptides that are critical for apoptosis.

The results of the Western blot analysis were expressed as the relative ratio of the specific compound band compared with the internal reference. Compound **4f** (3.24 µM) significantly increased the expression of the Bax protein and decreased that of Bcl-2 (Figure 4) in a time-dependent manner (3–24 h) that resulted in the induction of apoptosis. Caspase-3 expression increased by 134.3% in HeLa cells at 24 h after treatment with **4f** indicating the activation of apoptosis by the imidazole derivative.

These results suggest that compound **4f** induces apoptosis in HeLa cells by upregulating the mitochondrial apoptotic pathways. Apoptosis is an internal procedure developed by multicellular organisms to control cell proliferation. It is also a response to DNA damage during or after cell development [23]. Targeting apoptotic pathways in premalignant and malignant cells is an effective strategy for cancer prevention and treatment [24]. The intrinsic apoptotic pathway is initiated when an injury occurs within the cell and is activated by intracellular signals from mitochondria. The present results indicate that **4f** induced apoptosis through the mitochondrial pathway. Bcl-2 is an antiapoptotic protein that binds with Bax and Bak to protect against mitochondrial damage; the present results demonstrate that **4f** treatment decreased Bcl-2 that led to mitochondrial dysfunction and apoptosis [25]. An increasing amount of evidence shows that apoptosis plays an important role in eliminating cancer cells without damaging normal cells or surrounding tissues [26]. This may explain why normal cells were more tolerant of our compounds and why the compounds show a favorable selectivity index.

## 3. Material and Methods

### 3.1. Chemistry

Chromatography was conducted on silica gel 60, 100–200 mesh (Haiyang, Qingdao, China), under high pressure supplied by an aquarium air pump (SB-2800, Sobo, Guangzhou, China). Analytic thin-layer chromatography (TLC) was performed on precoated silica gel 60 F254 plates (Qingdao Haiyang, China). The plates were visualized using UV 254 nm light and/or iodine. Melting points (m.p.) were determined in open capillary tubes and were uncorrected. Chemical shifts are reported in ppm relative to tetramethyl silane. The structures of all the synthesized compounds were confirmed by ^1^H, ^13^C-NMR, and MS. NMR spectra were recorded on an AV-300 FT-NMR spectrometer (Bruker BioSpin AG, Billerica, MA, USA), and MS on an HP1100 LC/MSD electrospray ionization mass spectrometer (Agilent Technologies, Santa Clara, CA, USA). All the chemicals were of analytic grade. The detailed preparation procedures of intermediates, byproducts, and target compounds are described below.

### 3.2. 2-Ethyl-1-(4-nitrophenyl)-1H-imidazole *(**1**)*

In the presence of NaOH (0.40 g, 10 mmol, Sinapharm, Beijing China), a mixture of 4-fluoronitrobenzene (1.41 g, 10 mmol, Ouhe, Beijing, China) and 2-ethylimidazole (0.97 g, 10 mmol, Ouhe, Beijing, China) in dimethyl sulfoxide (DMSO, 30 mL, Sinapharm, Beijing China) was stirred at 90 °C for 15 h. The end of the reaction was confirmed by TLC. The reaction mixture was cooled and poured into iced cold water (50 mL) while stirring. The formed precipitate was filtered, dried, and gave a yellow solid in 94% yield.

### 3.3. 4-(2-Ethyl-1H-imidazol-1-yl)benzenamine *(**2**)*

Hydrogen was added to a solution of **1** (2.17 g, 10 mmol) and Pd/C (0.44 g, 10%, 60% H_2_O, Ouhe, Beijing, China) in absolute ethanol (30 mL, Sinapharm, Beijing China). The reaction mixture was heated to 70 °C while stirring for 12 h. TLC revealed the completion of the reaction. After the reaction mixture was cooled, the Pd/C was repeatedly removed by suction filtration, and evaporated to dryness in 98% yield.

### 3.4. 3-(4-(2-Ethyl-1H-imidazol-1-yl)phenylcarbamoyl)propanoic Acid *(**3**)*

To a mixture of compound **2** (1.87 g, 10 mmol) in toluene (30 mL, Sinapharm, Beijing China) was added succinic anhydride (1.00 g, 10 mmol, Ouhe, Beijing, China), the mixture was refluxed for 6 h. TLC revealed the completion of the reaction. After being cooled, the mixture was filtered and produced a white solid in 92% yield. m.p. 163–165 °C; ^1^H-NMR (300 MHz, CDCl_3_), *δ*: 1.08–1.13 (t, 3H, *J* = 15.0 Hz, CH_3_), 2.54–2.61 (m, 6H, (CH_2_)_3_), 6.91 (s, 1H, Ar-H), 7.20 (s, 1H, Ar-H), 7.32–7.35 (d, 2H, *J* = 9.0 Hz, Ar-H), 7.70–7.73 (d, 2H, *J* = 9.0 Hz, Ar-H), 10.20 (s, 1H, NH), 12.14 (s, 1H, OH).

### 3.5. General Procedure for the Preparation of Compounds *(**4a–4s**)*

A mixture of compound **3** (0.29 g, 1 mmol), organic amines (1 mmol, Ouhe, Beijing, China), DMAP (0.12 g, 1 mmol, Ouhe, Beijing, China), and EDCI (0.19 g, 1 mmol, Ouhe, Beijing, China) in *N,N*-dimethyl formamide (DMF, 20 mL, Sinapharm, Beijing China) was heated to 70 °C while stirring for 12 h. TLC revealed the completion of the reaction. After being cooled, the precipitate was filtered and washed with water. The crude products were dissolved in about 2 mL of dichloromethane and recrystallized with petroleum ether (about 20 mL).

#### 3.5.1. 1-(4-(3-(Propylcarbamoyl)propionamide)phenyl)-2-ethyl Imidazole (**4a**)

Yield 44%; m.p. 128–130 °C; ^1^H-NMR (300 MHz, (CD_3_)_2_SO), *δ*: 0.81–0.86 (t, 3H, *J* = 15.0 Hz, CH_3_), 1.08–1.13 (t, 3H, *J* = 15.0 Hz, CH_3_), 1.36–1.44 (m, 2H, CH_2_), 2.42–2.44 (t, 2H, *J* = 6.0 Hz, CH_2_), 2.53–2.61 (m, 4H, CH_2_-CH_2_), 2.97–3.03 (q, 2H, *J* = 18.0 Hz, CH_2_), 6.90 (s, 1H, Ar-H), 7.18 (s, 1H, Ar-H), 7.31–7.34 (d, 2H, *J* = 9.0 Hz, Ar-H), 7.70–7.73 (d, 2H, *J* = 9.0 Hz, Ar-H), 7.84 (s, 1H, -NH), 10.15 (s, 1H, -NH). ^13^C-NMR (100 MHz, (CD_3_)_2_SO), *δ*: 11.84, 12.68, 20.43, 22.85, 30.76, 32.28, 119.95, 121.38, 126.40, 127.39, 132.62, 139.53, 148.93, 171.25, 171.41. ESI-HRMS *m*/*z*, calculated for [C_18_H_2__5_N_4_O_2_] [M + 1] 329.1972, found 329.1966.

#### 3.5.2. 1-(4-(3-(Butylcarbamoyl)propionamide)phenyl)-2-ethyl Imidazol (**4b**)

Yield 53%; m.p. 112–115 °C; ^1^H-NMR (300 MHz, (CD_3_)_2_SO), *δ*: 0.83–0.88 (t, 3H, *J* = 15.0 Hz, CH_3_), 1.08–1.13 (t, 3H, *J* = 15.0 Hz, CH_3_), 1.25–1.37 (m, 4H, CH_2_-CH_2_), 2.41–2.43 (d, 2H, *J* = 9.0 Hz, CH_2_), 2.56–2.58 (m, 4H, CH_2_-CH_2_), 3.03–3.05 (d, 2H, *J* = 6.0 Hz, CH_2_). 6.90 (s, 1H, NH), 7.18 (s, 1H, Ar-H), 7.31–7.34 (d, 2H, *J* = 9.0 Hz, Ar-H), 7.70–7.73 (d, 2H, *J* = 9.0 Hz, Ar-H), 7.84 (s, 1H, Ar-H), 10.16 (s, 1H, NH). ^13^C-NMR (100 MHz, (CD_3_)_2_SO), *δ*: 12.68, 14.10, 19.99, 20.43, 30.78, 31.72, 32.29, 38.62, 119.93, 121.37, 126.39, 127.40, 139.53, 171.22, 171.31. ESI-HRMS *m*/*z*, calculated for [C_19_H_2__7_N_4_O_2_] [M + 1] 343.2129, found 343.2123.

#### 3.5.3. 1-(4-(3-(Amylcarbamoyl)propionamide)phenyl)-2-ethyl Imidazole (**4c**)

Yield 32%; m.p. 95–98 °C; ^1^H-NMR (300 MHz, (CD_3_)_2_SO), *δ*: 0.83–0.87 (t, 3H, *J* = 12.0 Hz, CH_3_), 1.08–1.13 (t, 3H, *J* = 15.0 Hz, CH_3_), 1.24–1.25 (d, 4H, *J* = 3.0 Hz, CH_2_-CH_2_), 1.36–1.41 (m, 2H, CH_2_), 2.39–2.43 (t, 2H, *J* = 12.0 Hz, CH_2_), 2.53–2.61 (m, 4H, CH_2_-CH_2_), 3.00–3.06 (q, 2H, CH_2_), 6.90 (s, 1H, Ar-H), 7.18 (s, 1H, Ar-H), 7.31–7.34 (d, 2H, *J* = 9.0 Hz, Ar-H), 7.70–7.73 (d, 2H, *J* = 9.0 Hz, Ar-H), 7.82 (s, 1H, NH), 10.14 (s, 1H, NH). ^13^C-NMR (100 MHz, (CD_3_)_2_SO), *δ*: 12.69, 14.35, 20.43, 22.30, 29.05, 29.29, 30.77, 32.30, 119.91, 121.38, 126.39, 127.40, 132.60, 139.53, 148.90, 171.24, 171.32. ESI-HRMS *m*/*z*, calculated for [C_20_H_29_N_4_O_2_] [M + 1] 357.2285, found 357.2279.

#### 3.5.4. 1-(4-(3-(Phenylcarbamoyl)propionamide)phenyl)-2-ethylimidazole (**4d**)

Yield 45%; m.p. 170–173 °C; ^1^H-NMR (300 MHz, (CD_3_)_2_SO), *δ*: 1.08–1.13 (t, 3H, *J* = 15.0 Hz, CH_3_), 2.53–2.61 (q, 2H, -CH_2_), 2.68 (m, 4H, CH_2_-CH_2_), 6.90 (s, 1H, Ar-H), 6.99–7.04 (t, 1H, *J* = 15.0 Hz, Ar-H), 7.18 (s, 1H, Ar-H), 7.26–7.34 (m, 4H, Ar-H), 7.58–7.60 (d, 2H, *J* = 6.0 Hz, Ar-H), 7.72–7.74 (d, 2H, *J* = 6.0 Hz, Ar-H), 10.00 (s, 1H, NH), 10.22 (s, 1H, NH). ^13^C-NMR (100 MHz, (CD_3_)_2_SO), *δ*: 12.68, 20.43, 31.62, 119.40, 119.95, 121.39, 123.37, 126.43, 127.41, 129.10, 132.65, 139.50, 139.76, 148.92, 170.74, 171.13. ESI-HRMS *m*/*z*, calculated for [C_21_H_2__3_N_4_O_2_] [M + 1] 363.1816, found 363.1809.

#### 3.5.5. 1-(4-(3-(4-Cl-Phenylcarbamoyl)propionamide)phenyl)-2-ethyl Imidazol (**4e**)

Yield 39%; m.p. 184–188 °C; ^1^H-NMR (300 MHz, (CD_3_)_2_SO), *δ*: 1.10–1.15 (t, 3H, *J* = 15.0 Hz, CH_3_), 2.55–2.58 (d, 2H, *J* = 9.0 Hz, CH_2_), 2.68 (m, 2H, CH_2_), 2.81 (m, 2H, CH_2_), 6.90–6.95 (d, 1H, *J* = 15.0 Hz, Ar-H), 7.18 (s, 1H, Ar-H), 7.31–7.34 (m, 4H, Ar-H), 7.41–7.44 (d, 1H, *J* = 9.0 Hz, Ar-H), 7.54–7.57 (d, 1H, *J* = 9.0 Hz, Ar-H), 7.61–7.64 (d, 1H, *J* = 9.0 Hz, Ar-H), 7.71–7.73 (d, 1H, *J* = 9.0 Hz, Ar-H), 10.15 (s, 1H, NH), 10.23 (s, 1H, NH). ^13^C-NMR (100 MHz, (CD_3_)_2_SO), *δ*: 12.62, 12.68, 20.43, 20.53, 28.97, 31.60, 99.98, 119.93, 120.90, 121.39, 126.34, 126.42, 128.56, 129.02, 132.65, 138.70, 139.47, 170.93, 171.05, 177.24. ESI-HRMS *m*/*z*, calculated for [C_21_H_2__2_ClN_4_O_2_] [M + 1] 397.1426, found 397.1421.

#### 3.5.6. 1-(4-(3-(4-Br-Phenylcarbamoyl)propionamide)phenyl)-2-ethyl Imidazol (**4f**)

Yield 32%; m.p. 215–218 °C; ^1^H-NMR (300 MHz, (CD_3_)_2_SO), *δ*: 1.08–1.13 (t, 3H, *J* = 15.0 Hz, CH_3_), 2.53–2.61 (q, 2H, CH_2_), 2.68 (s, 4H, CH_2_-CH_2_), 6.90 (s, 1H, Ar-H), 7.18 (s, 1H, Ar-H), 7.32–7.34 (d, 2H, *J* = 6.0 Hz, Ar-H), 7.45–7.48 (d, 2H, *J* = 9.0 Hz, Ar-H), 7.56–7.59 (d, 2H, *J* = 9.0 Hz, Ar-H), 7.71–7.74 (d, 2H, *J* = 9.0 Hz, Ar-H), 10.15 (s, 1H, NH), 10.23 (s, 1H, NH). ^13^C-NMR (100 MHz, (CD_3_)_2_SO), *δ*: 12.67, 20.42, 31.60, 119.94, 121.30, 126.41, 127.40, 131.92, 139.11, 139.45, 148.89, 170.96. ESI-HRMS *m*/*z*, calculated for [C_21_H_2__2_BrN_4_O_2_] [M + 1] 441.0921, found 441.0918.

#### 3.5.7. 1-(4-(3-(4-F-Phenylcarbamoyl)propionamide)phenyl)-2-ethyl Imidazol (**4g**)

Yield 59%; m.p. 75–78 °C; ^1^H-NMR (300 MHz, (CD_3_)_2_SO), *δ*: 1.08–1.13 (t, 3H, *J* = 15.0 Hz, CH_3_), 2.53–2.61 (q, 2H, CH_2_), 2.68–2.73 (d, 4H, *J* = 15.0 Hz, CH_2_-CH_2_), 6.90 (s, 1H, Ar-H), 7.10–7.18 (q, 3H, Ar-H), 7.31–7.34 (d, 2H, *J* = 9.0 Hz, Ar-H), 7.61 (s, 2H, Ar-H), 7.71–7.74 (d, 2H, *J* = 9.0 Hz, Ar-H), 10.06 (s, 1H, NH), 10.22 (s, 1H, NH). ^13^C-NMR (100 MHz, (CD_3_)_2_SO), *δ*: 12.68, 20.43, 31.50, 31.68, 115.50, 115.79, 119.94, 121.02, 121.12, 121.39, 126.42, 127.40, 132.64, 136.14, 139.48, 148.90, 170.64, 171.09. ESI-HRMS m/z, calculated for [C_21_H_2__2_FN_4_O_2_] [M + 1] 381.1721, found 381.1716.

#### 3.5.8. 1-(4-(3-(p-Xylenecarbamoyl)propionamide)phenyl)-2-ethyl Imidazole (**4h**)

Yield 53%; m.p. 200–202 °C; ^1^H-NMR (300 MHz, (CD_3_)_2_SO), *δ*: 1.08–1.13 (t, 3H, *J* = 15.0 Hz, CH_3_), 2.24 (s, 3H, CH_3_), 2.53–2.61 (q, 2H, CH_2_), 2.67 (s, 4H, CH_2_-CH_2_), 6.90 (s, 1H, Ar-H), 7.07–7.10 (d, 2H, *J* = 9.0 Hz, Ar-H), 7.18 (s, 1H, Ar-H), 7.31–7.34 (d, 2H, *J* = 9.0 Hz, Ar-H), 7.46–7.49 (d, 2H, *J* = 9.0 Hz, Ar-H), 7.71–7.74 (d, 2H, *J* = 9.0 Hz, Ar-H), 9.89 (s, 1H, NH), 10.20 (s, 1H, NH). ^13^C-NMR (100 MHz, (CD_3_)_2_SO), *δ*: 12.69, 20.43, 20.87, 31.57, 31.77, 119.40, 119.93, 121.40, 126.42, 127.38, 129.48, 132.20, 132.62, 137.27, 139.51, 148.90, 170.46, 171.15. ESI-HRMS *m*/*z*, calculated for [C_22_H_2__5_N_4_O_2_] [M + 1] 377.1972, found 377.1968.

#### 3.5.9. 1-(4-(3-(4-Methoxybenzecarbamoyl)propionamide)phenyl)-2-ethyl Imidazole (**4i**)

Yield 38%; m.p. 160–163 °C; ^1^H-NMR (300 MHz, (CD_3_)_2_SO), *δ*: 1.08–1.13 (t, 3H, *J* = 15.0 Hz, CH_3_), 2.53–2.66 (m, 6H, (CH_2_)_3_), 3.71 (s, 3H, CH_3_), 6.84–6.90 (t, 3H, J = 18.0 Hz, Ar-H), 7.18 (s, 1H, Ar-H), 7.31–7.34 (d, 2H, *J* = 9.0 Hz, Ar-H), 7.48–7.51 (d, 2H, *J* = 9.0 Hz, Ar-H), 7.71–7.74 (d, 2H, *J* = 9.0 Hz, Ar-H), 9.84 (s, 1H, NH), 10.20 (s, 1H, NH). ^13^C-NMR (100 MHz, (CD_3_)_2_SO), *δ*: 12.69, 20.43, 31.50, 31.85, 55.62, 114.27, 119.98, 120.96, 121.39, 126.42, 127.39, 132.64, 132.96, 139.51, 148.94, 155.47, 170.19, 171.18. ESI-HRMS *m*/*z*, calculated for [C_22_H_2__5_N_4_O_3_] [M + 1] 393.1921, found 393.1916.

#### 3.5.10. 1-(4-(3-(4-amino-Phenylcarbamoyl)propionamide)phenyl)-2-ethylimidazole (**4j**)

Yield 62%; m.p. 229–231 °C; ^1^H-NMR (300 MHz, (CD_3_)_2_SO), *δ*: 1.08–1.13 (t, 3H, *J* = 15.0 Hz, CH_3_), 2.53–2.65 (m, 6H, (CH_2_)_3_), 4.81 (s, 2H, NH_2_), 6.47–6.50 (d, 2H, *J* = 9.0 Hz, Ar-H), 6.90 (s, 1H, Ar-H), 7.19–7.23 (t, 3H, *J* = 12.0 Hz, Ar-H), 7.31–7.34 (d, 2H, *J* = 9.0 Hz, Ar-H), 7.71–7.74 (d, 2H, *J* = 9.0 Hz, Ar-H), 9.56 (s, 1H, NH), 10.20 (s, 1H, NH). ^13^C-NMR (100 MHz, (CD_3_)_2_SO), *δ*: 12.69, 20.43, 31.43, 114.25, 119.91, 121.15, 121.40, 126.41, 127.39, 129.03, 132.60, 139.52, 144.98, 148.89, 169.59, 171.24. ESI-HRMS *m*/*z*, calculated for [C_21_H_2__4_N_5_O_2_] [M + 1] 378.1925, found 378.1921.

#### 3.5.11. 1-(4-(3-(Benzylcarbamoyl)propionamide)phenyl)-2-ethyl Imidazol (**4k**)

Yield 53%; m.p. 123–126 °C; ^1^H-NMR (300 MHz, CDCl_3_), *δ*: 1.22–1.27 (t, 3H, *J* = 15.0 Hz, CH_3_), 2.64–2.96 (m, 8H, (CH_2_)_4_), 4.39–4.47 (m, 3H, Ar-H), 6.22 (s, 1H, Ar-H), 6.97 (s, 1H, Ar-H), 7.09 (s, 1H, NH), 7.30–7.38 (m, 4H, Ar-H), 7.66–7.69 (d, 2H, *J* = 9.0 Hz, Ar-H), 9.05 (s, 1H, NH). ^13^C-NMR (100 MHz, CDCl_3_), *δ*: 13.74, 22.07, 32.73, 33.95, 45.09, 122.62, 123.40, 128.15, 128.49, 129.08, 129.44, 130.45, 135.02, 140.95, 141.61, 174.09, 175.44. ESI-HRMS *m*/*z*, calculated for [C_22_H_2__5_N_4_O_2_] [M + 1] 377.1972, found 377.1968.

#### 3.5.12. 1-(4-(3-(4-Cl-Benzylcarbamoyl)propionamide)phenyl)-2-ethyl Imidazol (**4l**)

Yield 45%; m.p. 138–142 °C; ^1^H-NMR (300 MHz, (CD_3_)_2_SO), *δ*: 1.09–1.14 (t, 3H, *J* = 15.0 Hz, CH_3_), 2.53–2.62 (q, 6H, (CH_2_)_3_), 4.25–4.27 (d, 2H, *J* = 6.0 Hz, CH_2_), 6.91 (s, 1H, Ar-H), 7.19 (s, 1H, Ar-H), 7.26–7.28 (d, 2H, *J* = 6.0 Hz, Ar-H), 7.32–7.35 (q, 4H, Ar-H), 7.72–7.75 (d, 2H, *J* = 9.0 Hz, Ar-H), 8.46 (s, 1H, NH), 10.18 (s, 1H, NH). ^13^C-NMR (100 MHz, (CD_3_)_2_SO), *δ*: 12.69, 20.45, 30.75, 32.19, 41.86, 119.94, 121.39, 126.42, 127.41, 128.57, 129.40, 131.63, 132.64, 139.18, 139.53, 148.91, 171.17, 171.77. ESI-HRMS *m*/*z*, calculated for [C_22_H_2__4_ClN_4_O_2_] [M + 1] 411.1582, found 411.1578.

#### 3.5.13. 1-(4-(3-(4-Br-Benzylcarbamoyl)propionamide)phenyl)-2-ethylimidazole (**4m**)

Yield 53%; m.p. 143–145 °C; ^1^H-NMR (300 MHz, (CD_3_)_2_SO), *δ*: 1.08–1.13 (t, 3H, *J* = 15.0 Hz, CH_3_), 2.57–2.62 (t, 4H, CH_2_-CH_2_), 3.32 (s, 2H, CH_2_), 4.23–4.25 (d, 2H, *J* = 6.0 Hz, CH_2_), 6.90 (s, 1H, Ar-H), 7.19–7.22 (d, 3H, *J* = 9.0 Hz, Ar-H), 7.32–7.35 (d, 2H, *J* = 9.0 Hz, Ar-H), 7.45–7.47 (d, 2H, *J* = 6.0 Hz, Ar-H), 7.71–7.74 (d, 2H, J = 9.0 Hz, Ar-H), 8.45 (s, 1H, NH), 10.18 (s, 1H, NH). ^13^C-NMR (100 MHz, (CD_3_)_2_SO), *δ*: 12.69, 20.44, 41.90, 119.92, 121.38, 126.40, 127.40, 129.76, 131.47, 171.15, 171.74. ESI-HRMS *m*/*z*, calculated for [C_22_H_2__4_BrN_4_O_2_] [M + 1] 455.1077, found 455.1074.

#### 3.5.14. 1-(4-(3-(4-F-Benzylcarbamoyl)propionamide)phenyl)-2-ethylimidazole (**4n**)

Yield 57%; m.p. 137–140 °C; ^1^H-NMR (300 MHz, (CD_3_)_2_SO), *δ*: 1.08–1.13 (t, 3H, *J* = 15.0 Hz, CH_3_), 2.52–2.66 (q, 6H, (CH_2_)_3_), 4.25–4.27 (d, 2H, *J* = 6.0 Hz, CH_2_), 6.91 (s, 1H, Ar-H), 7.07–7.13 (t, 2H, Ar-H), 7.19 (s, 1H, Ar-H), 7.26–7.35 (q, 4H, Ar-H), 7.71–7.74 (d, 2H, J = 9.0 Hz, Ar-H), 8.43 (t, 1H, NH), 10.18 (s, 1H, NH). ^13^C-NMR (100 MHz, (CD_3_)_2_SO), *δ*: 12.68, 20.44, 30.73, 32.18, 41.80, 115.19, 115.47, 119.92, 121.39, 126.42, 127.41, 129.43, 129.54, 132.62, 136.24, 136.28, 139.52, 148.90, 159.94, 163.14, 171.18, 171.67. ESI-HRMS *m*/*z*, calculated for [C_22_H_2__4_FN_4_O_2_] [M + 1] 395.1878, found 395.1871.

#### 3.5.15. 1-(4-(3-(4-Methylbenzylcarbamoyl)propionamide)phenyl)-2-ethylimidazole (**4o**)

Yield 37%; m.p. 164–166 °C; ^1^H-NMR (300 MHz, (CD_3_)_2_SO), *δ*: 1.08–1.13 (t, 3H, *J* = 15.0 Hz, CH_3_), 2.26 (s, 3H, CH_3_), 2.54–2.62 (q, 6H, (CH_2_)_3_), 4.21–4.23 (d, 2H, *J* = 6.0 Hz, CH_2_), 6.90 (s, 1H, Ar-H), 7.10–7.15 (q, 4H, Ar-H), 7.18 (s, 1H, Ar-H), 7.32–7.35 (d, 2H, *J* = 9.0 Hz, Ar-H), 7.71–7.74 (d, 2H, *J* = 9.0 Hz, Ar-H), 8.35 (s, 1H, NH), 10.17 (s, 1H, NH). ^13^C-NMR (100 MHz, (CD_3_)_2_SO), *δ*: 12.69, 20.44, 21.09, 30.75, 32.23, 42.26, 119.93, 121.39, 126.40, 127.40, 127.57, 129.19, 132.61, 136.11, 136.99, 139.53, 148.90, 171.20, 171.55. ESI-HRMS *m*/*z*, calculated for [C_23_H_2__7_N_4_O_2_] [M + 1] 391.2129, found 391.2121.

#### 3.5.16. 1-(4-(3-(4-Methoxybenzylcarbamoyl)propanamide)phenyl)-2-ethylimidazole (**4p**)

Yield 24%; m.p. 168–171 °C; ^1^H-NMR (300 MHz, (CD_3_)_2_SO), *δ*: 1.09–1.14 (t, 3H, *J* = 15.0 Hz, CH_3_), 2.54–2.62 (m, 6H, (CH_2_)_3_), 3.72 (s, 3H, CH_3_), 4.19–4.21 (d, 2H, *J* = 6.0 Hz, CH_2_), 6.83–6.86 (d, 2H, *J* = 9.0 Hz, Ar-H), 6.91 (s, 1H, Ar-H), 7.16–7.19 (d, 3H, *J* = 9.0 Hz, Ar-H), 7.32–7.35 (d, 2H, *J* = 9.0 Hz, Ar-H), 7.72–7.75 (d, 2H, *J* = 9.0 Hz, Ar-H), 8.34 (s, 1H, NH), 10.19 (s, 1H, NH). ^13^C-NMR (100 MHz, (CD_3_)_2_SO), *δ*: 12.68, 20.43, 30.75, 32.23, 55.50, 114.08, 119.92, 121.39, 126.40, 127.40, 128.89, 131.97, 132.61, 139.53, 148.90, 158.58, 171.21, 171.48. ESI-HRMS *m*/*z*, calculated for [C_23_H_2__7_N_4_O_3_] [M + 1] 407.2078, found 407.2069.

#### 3.5.17. 1-(4-(3-(2-Chlorophenylcarbamoyl)propionamide)phenyl)-2-ethyl Imidazole (**4q**)

Yield 57%; m.p. 116–120 °C; ^1^H-NMR (300 MHz, (CD_3_)_2_SO), *δ*: 1.08–1.13 (t, 3H, *J* = 15.0 Hz, CH_3_), 2.56–2.64 (m, 6H, (CH_2_)_3_), 4.33–4.34 (d, 2H, *J* = 3.0 Hz, CH_2_), 6.90 (s, 1H, Ar-H), 7.18 (s, 1H, Ar-H), 7.25–7.28 (m, 2H, Ar-H), 7.32–7.35 (d, 3H, *J* = 9.0 Hz, Ar-H), 7.41–7.42 (m, 1H, Ar-H), 7.72–7.75 (d, 2H, *J* = 9.0 Hz, Ar-H), 8.44 (t, 1H, NH), 10.17 (s, 1H, NH). ^13^C-NMR (100 MHz, (CD_3_)_2_SO), *δ*: 12.69, 20.44, 30.66, 32.13, 119.95, 121.40, 126.42, 127.40, 127.52, 128.94, 129.16, 129.48, 132.39, 132.63, 136.87, 139.52, 148.93, 171.19, 171.97. ESI-HRMS *m*/*z*, calculated for [C_22_H_2__4_ClN_4_O_2_] [M + 1] 411.1582, found 411.1573.

#### 3.5.18. 1-(4-(3-(Pyridinemethylcarbamoyl)propionamide)phenyl)-2-ethyl Imidazole (**4r**)

Yield 44%; m.p. 118–121 °C; ^1^H-NMR (300 MHz, (CD_3_)_2_SO), *δ*: 1.08–1.13 (t, 3H, *J* = 15.0 Hz, CH_3_), 2.57–2.64 (q, 6H, (CH_2_)_3_), 4.36–4.37 (d, 2H, *J* = 3.0 Hz, CH_2_), 6.91 (s, 1H, NH), 7.19–7.35 (m, 5H, Ar-H), 7.68–7.75 (q, 3H, Ar-H), 8.48–8.53 (t, 2H, *J* = 15.0 Hz, Ar-H), 10.22 (s, 1H, NH). ^13^C-NMR (100 MHz, (CD_3_)_2_SO), *δ*: 12.70, 20.44, 30.71, 32.15, 44.66, 119.93, 121.28, 121.41, 122.43, 126.41, 127.40, 132.61, 137.04, 139.54, 148.90, 149.21, 159.25, 171.22, 171.91. ESI-HRMS *m*/*z*, calculated for [C_21_H_2__4_N_5_O_2_] [M + 1] 378.1925, found 378.1920.

#### 3.5.19. 1-(4-(3-(Thiazolcarbamoyl)propionamide)phenyl)-2-ethyl Imidazole (**4s**)

Yield 27%; m.p. 168–170 °C; ^1^H-NMR (300 MHz, (CD_3_)_2_SO), *δ*: 1.08–1.13 (t, 3H, *J* = 15.0 Hz, CH_3_), 2.56–2.58 (q, 2H, CH_2_), 2.74–2.76 (q, 4H, CH_2_-CH_2_), 6.90 (s, 1H, Ar-H), 7.18 (s, 2H, Ar-H), 7.31–7.34 (d, 2H, *J* = 9.0 Hz, Ar-H), 7.45 (d, 1H, *J* = 9.0 Hz, Ar-H), 7.70–7.73 (d, 2H, *J* = 9.0 Hz, Ar-H), 10.24 (s, 1H, NH), 12.13 (s, 1H, NH). ^13^C-NMR (100 MHz, (CD_3_)_2_SO), *δ*: 12.68, 20.43, 30.34, 31.26, 113.60, 119.97, 121.37, 126.43, 127.41, 132.70, 137.99, 139.43, 148.91, 158.45, 170.78, 170.88. ESI-HRMS *m*/*z*, calculated for [C_18_H_20_N_5_O_2_S] [M + 1] 370.1332, found 370.1326.

### 3.6. Anticancer Activity

#### 3.6.1. Cell Culture

The three cancer cell lines namely the human lung cancer cell line (A549), human cervical cancer cell line (HeLa), human gastric carcinoma cell line (SGC-7901), and normal liver cell line (L-02) were acquired from the American Type Culture Collection (Manassas, VA, USA). A549 and SGC-7901 cells were routinely cultured in RPMI-1640. In addition, HeLa and L-02 cells were routinely cultured in DMEM. Media was supplemented with 10% fetal bovine serum (FBS). Cells were maintained at subconfluency at 37 °C in humidified air containing 5% CO_2_. The cells were monitored daily and maintained at 80% cell density. All compounds tested were dissolved in DMSO and then diluted by the culture medium before the treatment of cultured cells.

#### 3.6.2. Cell Viability Assay

Cytotoxicity of the tested samples was measured against each cell line using the 3-(4,5-dimethylthiazol-2-yl)-2,5-diphenyltetrazolium bromide (MTT) cell viability assay (Maher et al., 2019). Cancer cells (A549, SGC-7901, and HeLa) and L-02 normal cells were harvested during the logarithmic growth phase. All cells were seeded in 96-well plates at 10^4^ cells/well and were treated with 5-fluorouracil (5-FU), methotrexate (MTX), and the tested samples for 24 h at final concentrations of 0.3, 1, 3, 10, 30, and 100 µmol/mL. Next, 20 µL of freshly prepared MTT (Solarbio) solution (5 mg/mL of MTT in PBS) was added to each well, and the cells were cultured for 4 h at 37 °C. The supernatants were removed and resolved with 100 µL of DMSO, and the cells were shocked for 10 min at 37 °C. The optical density of the samples was measured at 490 nm using a microplate luminometer. Cell viability was expressed as the percentage change in absorbance compared to the control values.

### 3.7. Cell Apoptosis Analysis

Apoptosis and necrosis were analyzed by Hoechst/propidium iodide (CA1120, Solarbio, Beijing, China) double staining [27]. HeLa cells were seeded in six-well plates (10^4^ cells/well) and treated with IC_50_ concentrations of 5-FU or one of the tested derivatives for 0, 3, 6, 12, and 24 h. Next, the cells were collected and washed once in phosphate-buffered saline (PBS). The cells were then resuspended in 195 µL of binding buffer, and dual staining was performed with 3 µL of Hoechst and 3 µL of PI. Next, the cells were incubated for 15 min at room temperature in the dark. Apoptotic cells were analyzed using the EVOS FL Auto Cell Imaging System (Thermo Fisher Scientific, Shanghai, China).

### 3.8. Western Blotting Analysis

For performing Western blot analysis, the cells were washed with PBS and lysis buffer (P0013B, Beyotime). After incubation on ice for 30 min, the homogenate was centrifuged at 12,000*g* for 15 min at 4 °C, and the supernatant was used for protein analysis. Protein concentration was determined using the BCA protein assay (P0012S, Beyotime, Beijing, China). Sodium dodecyl sulfate-polyacrylamide gel electrophoresis was performed using the Bio-Rad Mini-Protean Tetra System (Bio-Rad, Hercules, CA, US). The proteins on the gel were then transferred onto polyvinylidene fluoride membranes (PVDF, Millipore, Beijing, China) for 2 h at 200 mA and blocked for 2 h with 5% skim milk at room temperature to prevent nonspecific binding. The blots were probed with the indicated primary antibodies followed by horseradish peroxidase (HRP)-conjugated goat anti-rabbit or anti-mouse IgG (Beyotime). Rabbit anti-Bcl-2 and mouse anti-actin antibodies were purchased from Beyotime. Rabbit anti-Bax, anti-Caspass-3, and anti-PARP monoclonal antibodies were purchased from Cell Signaling Technology (Danvers, MA, USA) [28].

### 3.9. Statistical Analysis

This procedure was performed by a one-way analysis of variance in SPSS 19.0 software (IBM, Amonk, NY, USA), and the results are presented as the means ± SD. A value of *p* < 0.05 was assumed to indicate statistical significance.

## 4. Conclusions

In this study, we synthesized 19 novel imidazole derivatives, characterized their spectral data, and evaluated their anticancer activity against several cancer cell lines. Investigation of the anticancer mechanisms of our most active compound **4f** indicated that it induced apoptosis in HeLa cells by regulating the apoptotic protein pathways. Our findings suggest that our novel compounds might have utility in the treatment of human cancer. Further follow-up studies are required to confirm this potential.

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
