# Peer review of "Synthesis and Evaluation of the Antitumor Activity of Novel 1-(4-Substituted phenyl)-2-ethyl Imidazole Apoptosis Inducers In Vitro"

_molecules, 2020, doi:10.3390/molecules25184293_

Round 1

Reviewer 1 Report

This paper reports the synthesis and evaluation of the anti-tumor activity of some new imidazoles.  The chemical synthesis is routine but the new compounds do not have analytical or HRMS data which are required standards for publication.  I have no doubt that the compounds are correct, based on the supplementary NMR data, but I think HRMS data are needed.  Assuming these data can be provided I have some further comments on the manuscript.

  1. The parent imidazole 2 has already been published by the same group, but the reference is not quoted (Sun et al. Latin American Journal of Pharmacy, 2018, 37, 768).
  2. Yields of the amides 4 are generally much lower than would be expected.  Some comment would be helpful.
  3. Yields should be quoted only to the nearest integer.
  4. Scheme 1 should contain an indication of the R groups. Perhaps the Table 1 can be associated with the synthesis as well as the biological testing.
  5. In the experimental descriptions for compounds 4, the reacting amines should be mentioned to benefit clarity for the reader.

Reviewer 2 Report

The manuscript is a logical extension of previously reported studies (ref. 6, as well as, Eur. J. Med. Chem. 2013, 60, 451) in synthesis and biological activity of a series of imidazole derivatives. I recommend publication in Molecules after the minor corrections and modifications listed below.

  1. In the Figure 1 it is necessary to present structure of compound I since authors indicated (page 2, line 51) that this compound showed activity against lung and liver cancer cells.
  2. Reference Eur. J. Med. Chem. 2013, 60, 451 should be included in revised manuscript.
  3. In the Figure 2 (or Table 1) it is strongly recommended to report in vitro antitumor activity (IC50) of compound I.
  4. In the Scheme 1 it is necessary to specify yields of the synthesized compounds. Additionally, the yields of final compounds 4a-4s should be included in Scheme 1 or Table 1.
  5. It would be interesting to present a comparison of antitumor activity of compounds 4a-4s with antitumor activity of compound I. Did authors evaluate antitumor activity of compounds published in Eur. J. Med. Chem. 2013, 60, 451.

Round 2

Reviewer 1 Report

The improved paper is now acceptable for publication.